# Adherence barriers and interventions to improve ART adherence in Sub-Saharan African countries: A systematic review protocol

Amos Buh[1]*, Raywat Deonandan[1], James Gomes[1], Alison Krentel[2], Olanrewaju Oladimeji[3,4], Sanni Yaya[5,6]*

1 Interdisciplinary School of Health Sciences, University of Ottawa, Ottawa, Ontario, Canada, 2 School of Epidemiology and Public Health, University of Ottawa, Ottawa, Ontario, Canada, 3 Faculty of Health Sciences, Department of Public Health, Walter Sisulu University, Mthatha, Eastern Cape, South Africa, 4 Harvard T.H. Chan School of Public Health, Harvard University, Cambridge, Massachusetts, United States of America, 5 School of International Development and Global Studies, University of Ottawa, Ottawa, Ontario, Canada, 6 Faculty of Medicine, University of Parakou, Parakou, Benin

* sanni.yaya@gmail.com (SY); abuh020@uottawa.ca (AB)

**Data Availability Statement:** Deidentified research data will be made publicly available when the study is completed and published.

## Abstract

### Background

The HIV/AIDS pandemic continues to be a major public health concern, particularly in Sub-Saharan Africa (SSA). Despite efforts to reduce new infections and deaths with the use of antiretroviral therapy (ART), SSA countries continue to bear the heaviest burden of HIV/AIDS globally, accounting for two-thirds of global new infections. The goal of this review is to identify common barriers to ART adherence as well as common effective interventions that can be implemented across SSA countries to improve ART adherence.

### Methods

A systematic review of published studies on adult HIV-positive patients aged 15 or above, that have assessed the barriers to ART adherence and interventions improving patients' adherence to ART in SSA countries shall be conducted. We will conduct electronic searches for articles that have been published starting from January 2010 onwards. The databases that shall be searched will include Medline Ovid, CINAHL, Embase, and Scopus. The review will include experimental and quasi-experimental studies such as randomized and non-randomized controlled trials as well as comparative before and after studies, and observational studies—cross-sectional studies, cohort studies, prospective and retrospective studies. Two independent reviewers will screen all identified studies, extract data and appraise the methodological quality of the studies using standard critical appraisal tools from the Joanna Briggs Institute. The extracted data will be subjected to a meta-analysis and narrative synthesis.

### Discussion

This review will synthesize existing evidence on ART adherence barriers and strategies for improving patient adherence to ART in SSA countries. It will identify common barriers to

**Funding:** The author(s) received no specific funding for this work.

**Competing interests:** The authors have declared that no competing interests exist.

**Abbreviations:** **AIDS**, Acquired Immune deficiency syndrome; **ART**, Antiretroviral Therapy; **CI**, Confidence interval; **GRADE**, Grading of Recommendations Assessment, Development and Evaluation; **HIV**, Human immunodeficiency virus; **JBI**, Joanna Briggs Institute; **JBI-MASTARI**, Joanna Briggs Institute Meta-Analysis of Statistics Assessment and Review instrum; **LMIC**, Low and Middle-Income Countries; **OR**, Odds Ratio; **PLWH**, People living with HIV; **PLWHA**, People living with HIV/AIDS; **PRISMA-P**, Preferred Reporting Items for Systematic Reviews and Meta-Analyses Protocols; **PROSPERO**, International Prospective Register of Systematic Reviews; **SSA**, Sub-Saharan Africa.

adherence and common interventions proven to improve adherence across SSA. We antici-pate that the findings of this review will provide information policy makers and stakeholders involved in the fight against HIV, will find useful in deriving better ways of not only retaining patients on treatment but having them adhere to their treatment.

## Review registration

This protocol has been registered with the International Prospective Register of Systematic Reviews (PROSPERO); registration number CRD42021262256.

## Background

The HIV/AIDS pandemic remains a significant global public health issue. From the beginning of the epidemic to the end of 2019, 75.7 million people (55.9 million– 100 million) have been infected with HIV [1]. In 2019, an estimated 38.0 million people worldwide were living with HIV, 1.7 million new infections occurred, and 690,000 (500,000–970,000) deaths from AIDS-related illnesses were reported [1,2]. Despite efforts to reduce new infections and deaths, Sub-Saharan Africa (SSA) bears the world's heaviest burden of HIV and AIDS, accounting for two-thirds of global new infections and 25.7 million HIV-positive people [3].

Evidence suggests that once people are diagnosed with HIV, it is critical to ensure that they are effectively linked to care [4]. At the moment, no cure has been discovered, and anti-retrovi-ral therapy (ART) is the only treatment modality available to prolong life and improve the quality of life of people living with HIV/AIDS (PLWHA) [5,6]. ART has completely changed the course of the HIV disease, transforming it from a potentially fatal infection to a manage-able chronic condition [7]. ART prevents further virus replication or multiplication, lowers the patient's viral load, raises CD4 counts, lowers the patient's risk of opportunistic infections and hospitalizations, improves the patient's quality of life, and lowers mortality [8–10]. The increase in the patient's CD4 cell count allows the individual's immune system to recover and produce more CD4 cells which fights off infections and other HIV-related cancers [1]. To be effective, ARTs has to be taken life-long as prescribed, with 100% adherence [2,3]. Adherence to ART reduces the viral load in an individual's body, prevents treatment failure and the likeli-hood of the emergence of drug-resistant strains of the virus and also prevents further transmis-sion of the virus to non-infected persons [1,4–6]. Poor or non-adherence–not taking ART every day and exactly as prescribed, can lead to drug resistance and treatment failure [1].

Although ART enables immune recovery and improves survival in PLWH [5], most health-care systems in SSA face numerous challenges emanating from scaling up ART for PLWHA. The most prominent of these challenges include suboptimal adherence to ART, poor retention in care and congestion of primary health care facilities [7]. These challenges might contribute to the continuous observance of non-adherence to ART in SSA countries where the burden of the disease is high [8]. Some studies have reported that the patient, disease, therapy and rela-tionship of patients with healthcare providers are associated with ART adherence issues [9,10]. Other studies have identified that, the major barriers to ART adherence include stigma, nega-tive perception, lack of family and community support [11,12]. Other factors that have been reported as common barriers to ART adherence include disclosure of status, unemployment, a lack of transportation to the health facility for ART, insufficient feeding, inadequate follow-ups, a lack of patient confidentiality, disability grants, and alternative forms of therapy [12]. Physical, economic, and emotional stress, travel away from home, business with other things,

depression, alcohol or drug use, and ART dosing frequency have all been identified as barriers to adherence [13,14]. Nonetheless, evidence on the specific barriers to ART adherence that are common in all SSA countries is limited.

With regards to ART adherence interventions, it has been reported that patients on ART face multiple barriers to adherence, and no single intervention is sufficient to ensure optimal adherence. As a result, health providers should consider a series of approaches that first identify patients at risk of poor adherence and then seek to establish the support needed to overcome adherence barriers [15]. Home-based nursing of patients on treatment, peer support for adolescents, and the prescription of LPV/r-containing regimens are among the interventions [16]. Another study found that cognitive behavioral interventions, education, treatment supporters, directly observed therapy, and active adherence reminder devices (such as mobile phone text messages) can help patients stick to their ART regimen [17]. However, there is a scarcity of literature on common specific interventions that are effective in improving adherence to ART in SSA countries.

Given that the HIV burden is disproportionately concentrated in SSA countries, which frequently have underdeveloped healthcare systems and limited resources for HIV treatment. Reviewing common barriers to ART adherence and effective interventions proven to improve ART adherence in these countries could assist countries and organizations involved in the fight against HIV in SSA in developing management policies that can be implemented in these countries to quickly curb the pandemic. As a result, the purpose of this review is to answer the question, "What are the common barriers to ART adherence, and what are the common effective interventions that can be implemented to improve ART adherence across SSA countries?"

## Methods

### Study design

This will be a systematic review of published studies that examined the barriers to ART adherence and interventions that improved patients' adherence to ART in resource-limited settings (SSA countries). The Preferred Reporting Items for Systematic Reviews and Meta-Analyses Protocols (PRISMA-P) criteria (Supplemental file 1) were used to develop this protocol [18]. We will use the PRISMA diagram (Supplemental file 2) to show how studies for this review shall be selected.

### Inclusion criteria

To select appropriate studies for this review, we will use the PICO (population, intervention, comparator and outcome) criteria. This will allow us to find and select the right studies that can address our research questions. Our inclusion criteria will therefore comprise:

### Population

This review will include studies conducted between 2010 and 2022 on adult HIV-positive patients aged 15 or above in SSA.

### Intervention

All studies that assessed the barriers to ART adherence and or evaluated interventions aimed at improving ART adherence among adults PLWH in SSA shall be included in this review.

### Comparator

The interventions may be in comparison with other strategies to identify the most effective and/or may be in comparison to no strategies/interventions (regular basic management).

## Outcomes

The review will include studies that assessed the following outcomes:

Primary outcome: Proportion of patients adhering to treatment following implementation of specific strategies.

Secondary outcomes: Proportion of patients retained in care, prevalence of opportunistic infections and or the worsening/severity of the patient's stage of HIV infection following specific treatment interventions. Included studies must measure viral load and CD4 cell counts as an indication of the treatment adherence and efficacy.

## Types of studies

This review will include experimental and quasi-experimental studies such as studies with controlled interventions that have assessed barriers to ART adherence and interventions improving ART adherence in SSA countries. These will include randomized and non-randomized controlled trials as well as comparative before and after studies, and observational studies (cross-sectional studies, cohort studies, prospective and retrospective studies). Only studies conducted between 2010 and 2022, in SSA, and involving adults 15 years and above shall be included in this review.

## Language

Only studies written in English and or French will be included in this systematic review.

## Search strategy

A three-step strategy shall be used to find published studies on barriers to ART adherence and interventions improving adherence to ART among adult PLWH in SSA. An initial search through the Medline Ovid database shall first be conducted using an analysis of text words found in the title and abstract, and the index terms used in describing the article. Secondly, keywords and index terms shall be identified to search for studies in selected databases. Finally, additional studies not found in the databases shall be looked for using the reference list of selected studies from the first and second searches.

For this review, the databases that shall be searched will include Medline Ovid, CINAHL, Embase, and Scopus. We will also use search engines and directories such as Google scholar and Centres for Disease Control and Prevention (CDC) to search for unpublished studies.

Some of the keywords we have used for our initial searches in Medline Ovid include *"ART"*, *"adherence"*, *"retention in care"*, *"non-adherence"*, *"barriers to ART adherence"*, *"adherence strategies"*, *"HIV"*, *"adults PLWH"*, *"sub-Saharan African countries"* (Supplementary file 3).

## Study screening and selection

Studies that shall be identified in searched databases shall be saved in Zotero and exported to the Covidence software for screening. The inclusion and exclusion criteria of this study shall also be imported to the Covidence software and the software will be used for title, abstract and full-text screening. After importing references and inclusion/exclusion criteria into the software, two independent reviewers will screen titles of included studies following the eligible criteria. All conflicts between the two reviewers will be resolved either through discussion or by a third reviewer. This same procedure will be applied for abstract screening. After the abstract screening, full texts of potentially eligible studies shall be retrieved and independently assessed for eligibility by two reviewers. Any conflicts or disagreement between the two reviewers over the eligibility of a given study shall be resolve in a similar manner as for the title and abstract screening.

## Assessment of methodological quality

Two independent reviewers shall be used to assess the methodological validity of the studies that shall be selected for retrieval using standard critical appraisal tools from the Joanna Briggs Institute for Meta-Analysis of Statistics Assessment and Review Instrument (JBI-MAStARI) (Supplementary file 4). Any disagreement between the two reviewers shall be settled through discussions or by a third reviewer.

## Data extraction

Data shall be extracted from selected studies using a standardized data extraction tool from the Joanna Briggs Institute Meta-Analysis of Statistics Assessment and Review instrument (Supplementary file 5). The extracted data shall include specific details about the barriers to ART adherence, strategies or interventions improving ART adherence, study populations, study methods and outcomes significant to the review question and objective. In the event of any missing data from a study, the corresponding author of the study shall be contacted to provide the missing data. The data shall be independently extracted by two reviewers.

## Data synthesis

If we end up with many studies (10 or more), we will conduct both a meta-analysis and a narrative synthesis of the various interventions improving ART adherence. The meta-analysis will be done to identify the interventions with a significant impact in improving patients' adherence to ART. However, if we end up with few studies, we will just do a narrative synthesis of the barriers to ART adherence and the interventions improving adherence in SSA.

For the meta-analysis, we will first assess the statistical heterogeneity with $I^2$, which indicates the percentage of the total variation across studies; where 0% - 40% indicates low heterogeneity, 30% - 60% indicates moderate heterogeneity, 50% - 90% indicates substantial heterogeneity, and 75% - 100% indicates considerable heterogeneity. If there is a substantial amount of heterogeneity (75%), then sources of heterogeneity will be examined through subgroup and sensitivity analyses. We will also use Chi-square test to test the heterogeneity and consider P-values < 0.05 as statistically significant. We will select a fixed-effects model for significant homogeneous studies; otherwise, we will apply a random-effects model. All outcomes will be summarized using odds ratios (OR) and 95% confidence intervals (CI). An OR<1 will indicate a lower rate of outcome among the group of patients who were treated following a given intervention. Publication bias will be assessed by visual inspections of funnel plots and Egger's test.

For the narrative synthesis, we will describe the barriers to ART adherence and the interventions promoting adherence to ART, following Popay's guidance on the conduct of a narrative synthesis [19]. The narrative synthesis shall be structured by describing the studies following the barriers assessed and the type of strategies used to improve ART adherence. The identified barriers to ART adherence and the interventions improving adherence shall first be presented in a table followed by detailed description of each ART adherence barrier and intervention promoting treatment adherence.

## Confidence in cumulative evidence

The quality of evidence that shall be used in this review will be assessed by the Grades of Recommendation, Assessment, Development and Evaluation (GRADE) [20].

## Discussion

This review will synthesize existing evidence on ART adherence barriers and strategies for improving patient adherence to ART in SSA countries. We acknowledge that other reviews have been done in this area although with some limitations—some of the reviews are a little old, some have limitations in methods, and some included only articles published in English. For instance, in 2008, a review was published on barriers to access to antiretroviral treatment in developing countries. This review included articles published between 1996 and 2007 and only included articles published in English from the PubMed, FamMed and Cochrane databases [21]. In 2012, another review on adherence to ART in Cameroon described the trend in adherence rates and factors associated with adherence. This review included only studies conducted in one SSA country—Cameroon from January 1999 to May 2012 [22]. In yet another study assessing the effectiveness of treatment supporter interventions in ART adherence in sub-Saharan Africa, only studies published in English from 2003 to February 2019 were included in the review [23]. Finally, another review on interventions to improve adherence in sub-Saharan Africa only included published articles from the PubMed, Medline and Google Scholar databases [24]. All these reviews likely missed studies not found in the few accessed databases and those published in French.

In view of the limitations of the above published reviews, this review will include articles published in SSA countries from 2010 onward, in both English and French, and searches will be done in more databases than previous studies. It will identify recent evidence on common adherence barriers and common interventions that have been shown to improve adherence across SSA. We anticipate that the findings of this review will provide information that policymakers and stakeholders involved in the HIV fight will find useful in devising better ways to help patients adhere to ART. This protocol is an original systematic review protocol; any changes must be documented and updated in PROSPERO, as well as stated in the final review manuscript. The review findings will be published in a peer-reviewed journal at the conclusion of this study.

## Supporting information

**S1 File. PRISMA-P 2015 checklist.**
(DOCX)

**S2 File. PRISMA 2009 flow diagram.**
(DOCX)

**S3 File. Medline (Ovid) search strategies and results.**
(DOCX)

**S4 File. Joanna briggs institute for meta-analysis of statistics assessment and review instruments.**
(DOCX)

**S5 File. Data extraction form for quantitative research.**
(DOCX)

## Acknowledgments

We would like to thank Valentina Ly (the librarian) for guidance in developing the searching strategies.

## Declarations

### Consent for publication

Data that shall be used in this review shall be available in the public domain and as such, consent to publish is not needed. Also, we will not use any details, images or videos related to individual participants in the review.

## Author Contributions

**Conceptualization:** Amos Buh, Sanni Yaya.

**Formal analysis:** Sanni Yaya.

**Investigation:** Sanni Yaya.

**Methodology:** Amos Buh, Raywat Deonandan, James Gomes, Alison Krentel, Olanrewaju Oladimeji, Sanni Yaya.

**Project administration:** Sanni Yaya.

**Supervision:** Sanni Yaya.

**Writing – original draft:** Amos Buh, Sanni Yaya.

**Writing – review & editing:** Amos Buh, Raywat Deonandan, James Gomes, Alison Krentel, Olanrewaju Oladimeji, Sanni Yaya.

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
