## [Decision Letter · Decision Letter 0]

24 Apr 2022

PONE-D-22-05384Adherence barriers and interventions to improve ART adherence in Sub-Saharan African countries: a systematic review protocolPLOS ONE

Dear Dr. Yaya,

Thank you for submitting your manuscript to PLOS ONE. After careful consideration, we feel that it has merit but does not fully meet PLOS ONE’s publication criteria as it currently stands. Therefore, we invite you to submit a revised version of the manuscript that addresses the points raised during the review process.

In particular, the protocol proposes two reviews - one on barriers to adherence; and another on interventions to address adherence. Both these reviews have been undertaken by previous researchers; but none of these reviews are referenced in the manuscript. As such a convincing rationale for the added benefit of the proposed review to existing literature is not provided.

We look forward to receiving your revised manuscript.

Kind regards,

Isabelle Chemin, PhD

Academic Editor

PLOS ONE

Journal Requirements:

Reviewers' comments:

Reviewer's Responses to Questions

**Comments to the Author**

1. Does the manuscript provide a valid rationale for the proposed study, with clearly identified and justified research questions?

Reviewer #1: Yes

Reviewer #2: No

2. Is the protocol technically sound and planned in a manner that will lead to a meaningful outcome and allow testing the stated hypotheses?

Reviewer #1: Yes

Reviewer #2: No

3. Is the methodology feasible and described in sufficient detail to allow the work to be replicable?

Reviewer #1: Yes

Reviewer #2: No

4. Have the authors described where all data underlying the findings will be made available when the study is complete?

Reviewer #1: Yes

Reviewer #2: Yes

5. Is the manuscript presented in an intelligible fashion and written in standard English?

Reviewer #1: No

Reviewer #2: Yes

6. Review Comments to the Author

You may also provide optional suggestions and comments to authors that they might find helpful in planning their study.

Reviewer #1: The protocol is generally well drafted and will be valuable input to policy makers working on the area of HIV/AIDS and/or ART in the sub-saharan African countries. Providing concrete and cumulative evidence on barriers to and strategies for ART adherence will also be highly recommended to maximize treatment outcome in the region.

Having said this, I do have few concerns which require due attention while conducting this systematic review and meta-analysis

• Grammatical, syntax and technical problems

Abstract

o E.g. Background: Despite efforts to prevent new infections and deaths through antiretroviral therapy (ART)…… Can we prevent new infections with ART? Please rephrase the statement.

o Method: The review will include experimental and quasi-experimental studies such as…….. Please make it concise

o Why do you focus on articles published after 2010?

o The role of case studies and case-control studies for this protocol (interventional approach) should be clarified. It will be better if you include studies with interventional or observational nature (cohort or cross-sectional) ……

o Discussion: this review will provide information to policy makers….. better ways of not only retaining patients on treatment but also having them adhere to their treatment……

Method:

Search strategy requires refinement

• The keywords should be concise and explicit. It is not recommended to use phrases in searching unless it is a medical term or compulsory otherwise.

• Use Medical Subject Headings (MesH) during searching in major databases such as MEDLINE, PubMed. E.g. instead of using sub-saharan African Countries, apply “African, south of the Sahara” to address all countries in the region at once with MeSH.

• You can use truncation (*) to expand the search results as well. E.g. instead of using adulthood, apply adult*

• Make sure that you have prepared a ‘PICO’ tree for effectively addressing all studies conducted in the region.

• Google scholar and CDC are not databases, they are either search engines or directories…..

Discussion:

• The discussion is actually shallow. You can explain the rationale of conducting this protocol supported with citations.

• This protocol is an original research protocol??????? It is a systematic review protocol

Reviewer #2: The protocol proposes two reviews - one on barriers to adherence; and another on interventions to address adherence. Both these reviews have been undertaken by previous researchers; but none of these reviews are referenced in the manuscript. As such a convincing rationale for the added benefit of the proposed review to existing literature is not provided.

7. PLOS authors have the option to publish the peer review history of their article (what does this mean?). If published, this will include your full peer review and any attached files.

Reviewer #1: **Yes: **Mekonnen Sisay

Reviewer #2: **Yes: **Brian van Wyk

---

## [Author Response · Author response to Decision Letter 0]

2 May 2022

Review Comments to the Author

Reviewer #1: The protocol is generally well drafted and will be valuable input to policy makers working on the area of HIV/AIDS and/or ART in the sub-Saharan African countries. Providing concrete and cumulative evidence on barriers to and strategies for ART adherence will also be highly recommended to maximize treatment outcome in the region.

Having said this, I do have few concerns which require due attention while conducting this systematic review and meta-analysis

• Grammatical, syntax and technical problems

Abstract

o E.g. Background: Despite efforts to prevent new infections and deaths through antiretroviral therapy (ART)…… Can we prevent new infections with ART? Please rephrase the statement.

RESPONSE: Thank you for this observation and recommendation. The statement has been rephrased and now reads “Despite efforts to reduce new infections and deaths with the use of antiretroviral therapy (ART)…”.

o Method: The review will include experimental and quasi-experimental studies such as…….. Please make it concise

RESPONSE: This has been revised and now reads “The review will include experimental and quasi experimental studies such as studies with controlled interventions that have assessed barriers to ART adherence and interventions improving ART adherence in SSA countries.

o Why do you focus on articles published after 2010?

RESPONSE: Thanks for this relevant question. We acknowledge that the core concept of systematic reviews is to provide an overview of a specific topic through a systematic approach - identifying all potential evidence on a given topic, selecting the evidence objectively, assessing the evidence critically, and synthesizing the evidence reasonably. As such, the priority in a systematic review is to identify and collect every possible information regarding a topic; that is, collecting as many studies as possible. Thus, it is always better not to limit the search strategy in a systematic review without a good reason. With regards to our study however, we would like to point out that there was initially considerable reluctance to provide ART in developing countries due to concerns that treatment was too expensive, too complex, and that drug resistance would be promoted by inadequate programmes. Because of these concerns, treatment programmes began to deliver ART at scale only in the 2000s. Widespread access to affordable antiretrovirals became feasible after the announcement by an Indian generics manufacturer in early 2001 that triple therapy could be manufactured for less than a dollar a day. Given this awareness, our objective in this study is not to look far back into old evidence but to look at recent evidence that speaks directly to the current or near current experiences of people still living with HIV in SSA - reason for the restriction on the time period.

o The role of case studies and case-control studies for this protocol (interventional approach) should be clarified. It will be better if you include studies with interventional or observational nature (cohort or cross-sectional) ……

RESPONSE: Thank you for this observation and recommendation. We accept your recommendation to include interventional or observational studies such as cohort or cross-sectional studies. This section has been revised on the manuscript with tracked changes. 

o Discussion: this review will provide information to policy makers….. better ways of not only retaining patients on treatment but also having them adhere to their treatment……

RESPONSE: Thanks for this keen observation; the discussion section has been revised on the manuscript.

Method:

Search strategy requires refinement

• The keywords should be concise and explicit. It is not recommended to use phrases in searching unless it is a medical term or compulsory otherwise.

RESPONSE: Thanks for this recommendation; we reviewed our search strategy with the librarian, and this was taken into consideration. However, if any phrase has been used, it is to maximise the chance of not missing any article and where this is used, there was no other way than use the phrase.

• Use Medical Subject Headings (MesH) during searching in major databases such as MEDLINE, PubMed. E.g. instead of using sub-saharan African Countries, apply “African, south of the Sahara” to address all countries in the region at once with MeSH.

RESPONSE: Thank you for this suggestion. We reviewed our search strategy with the librarian, and this was taken into consideration; for example, line 34 of the attached search strategy reads “exp Africa South of the Sahara/”.

• You can use truncation (*) to expand the search results as well. E.g. instead of using adulthood, apply adult*

RESPONSE: Thanks for this suggestion; we consulted with the librarian and this was considered in our search strategy. Lines 35 to 53 of our search strategy has extensively used truncation to ensure we captured all articles e.g. cent* on line 35, adher* on line 43, complian* on line 46 etc.

• Make sure that you have prepared a ‘PICO’ tree for effectively addressing all studies conducted in the region.

RESPONSE: Thanks so much for this suggestion. We have described our inclusion criteria under methods following the ‘PICO’ criteria. When addressing all studies we will find through our searches of articles published in this region, we will adhere strictly to these criteria to ensure we capture all relevant studies for this review.

• Google scholar and CDC are not databases, they are either search engines or directories…..

RESPONSE: Thanks for this observation; revisions and correction on this section has been made on the manuscript.

Discussion:

• The discussion is actually shallow. You can explain the rationale of conducting this protocol supported with citations.

RESPONSE: Thanks for this suggestion; the discussion section has been revised following suggestions on the manuscript.

• This protocol is an original research protocol??????? It is a systematic review protocol

RESPONSE: Thank you for this observation; correction has been effected on the manuscript.

Reviewer #2: The protocol proposes two reviews - one on barriers to adherence; and another on interventions to address adherence. Both these reviews have been undertaken by previous researchers; but none of these reviews are referenced in the manuscript. As such a convincing rationale for the added benefit of the proposed review to existing literature is not provided.

RESPONSE: Thank you for this observation. We have now revised the discussion section of the manuscript and cited some previous reviews on this topic to support our rationale for this review. Tracked changes have been used to highlight all revisions done on the manuscript.

---

## [Editor Report · Decision Letter 1]

18 May 2022

Adherence barriers and interventions to improve ART adherence in Sub-Saharan African countries: a systematic review protocol

PONE-D-22-05384R1

Dear Dr. Yaya,

We’re pleased to inform you that your manuscript has been judged scientifically suitable for publication and will be formally accepted for publication once it meets all outstanding technical requirements.

Kind regards,

Isabelle Chemin, PhD

Academic Editor

PLOS ONE
---

## [Editor Report · Acceptance letter]

25 May 2022

PONE-D-22-05384R1 

Adherence barriers and interventions to improve ART adherence in Sub-Saharan African countries: a systematic review protocol 

Dear Dr. Yaya:

I'm pleased to inform you that your manuscript has been deemed suitable for publication in PLOS ONE. Congratulations! Your manuscript is now with our production department. 

Kind regards, 

on behalf of

Mrs Isabelle Chemin 

Academic Editor

PLOS ONE